# Pollution Indicators and HAB-Associated Halophilic Bacteria Alongside Harmful Cyanobacteria in the Largest Mussel Cultivation Area in Greece

**DOI:** 10.3390/ijerph19095285

**Published:** 2022-04-26

**Authors:** Maria P. Kalaitzidou, Maria V. Alvanou, Konstantinos V. Papageorgiou, Athanasios Lattos, Marina Sofia, Spyridon K. Kritas, Evanthia Petridou, Ioannis A. Giantsis

**Affiliations:** 1National Reference Laboratory for Marine Biotoxins, Department of Food Microbiology, Biochemical Control, Residues, Marine Biotoxins and Other Water Toxins, Directorate of Veterinary Center of Thessaloniki, Ministry of Rural Development and Food, 54627 Thessaloniki, Greece; mkalaitzidou1@gmail.com; 2Department of Animal Science, Faculty of Agricultural Sciences, University of Western Macedonia, 53100 Florina, Greece; mariaalvanou7@gmail.com (M.V.A.); lattosad@bio.auth.gr (A.L.); 33rd Military Veterinary Hospital, General Staff, Hellenic Ministry of Defense, 15th Km Thessaloniki-Vasilika, 57001 Thessaloniki, Greece; pgkostas@yahoo.gr; 4Faculty of Veterinary Science, University of Thessaly, 43100 Karditsa, Greece; msofia@uth.gr; 5Laboratory of Microbiology and Infectious Diseases, School of Veterinary Medicine, Aristotle University of Thessaloniki, 54124 Thessaloniki, Greece; skritas@vet.auth.gr (S.K.K.); epetri@vet.auth.gr (E.P.)

**Keywords:** seawater, bacterial communities, algal bloom, pollution, mussel, *Mytilus galloprovincialis*, contamination, biomarkers, public health, phylogeny

## Abstract

Taking into consideration the essential contribution of *Mytilus galloprovincialis* farming, it is of rising importance to add knowledge regarding bacterial species occurrence in water samples from aquaculture zones from the point of view of both the organism and public health. In the present study, we investigated the bacterial community existing in water samples from six *Mytilus galloprovincialis* aquaculture areas in the Thermaikos gulf, northern Greece, that may provoke toxicity in aquatic organisms and humans and may indicate environmental pollution in mussel production as well as algal blooms. Bacterial species were identified molecularly by sequencing of a partial 16s rRNA segment and were analyzed phylogenetically for the confirmation of the bacterial taxonomy. The results obtained revealed the presence of four bacterial genera (*Halomonas* sp., *Planococcus* sp., *Sulfitobacter* sp., and *Synechocystis* sp.). Members of the *Halomonas* and *Sulfitobacter* genera have been isolated from highly polluted sites, *Planococcus* bacteria have been identified in samples derived directly from plastic debris, and *Synechocystis* bacteria are in line with microcystin detection. In this context, the monitoring of the bacteria community in mussel aquaculture water samples from the Thermaikos gulf, the largest mussel cultivation area in Greece, represents an indicator of water pollution, microplastics presence, algal blooms, and toxin presence.

## 1. Introduction

Bacterial community alterations caused by marine pollution on account of various sources, some of which are plastics, microplastics, and nanoplastics, exhibit adverse effects by deteriorating the welfare of aquatic organisms and environment services [1]. Microplastics are water-insoluble solid polymer particles that are ≤5 mm in size, while nanoplastics are particles ≤1 μm in size [2]. Micro- and nanoplastics can be easily ingested by aquatic organisms and can thus be responsible for severe effects on organisms’ growth, survival, development, and reproduction [3]. It is estimated that a minimum of 5.25 trillion particles with a total weight of 268,940 tons of plastic items are floating in the world’s marine ecosystems across all five subtropical gyres, including the Mediterranean Sea [4].

Because of their characteristics, such as their small length and distribution, microplastics are easily colonized by bacteria, creating biofilms [5] However, in some cases, the microbial communities inhabiting plastics have been found to be different from those in the water samples around them [6,7,8]. In studies isolating bacteria from polluted sites, such as landfill sites, it has been shown that these bacteria belonged to Proteobacteria, Firmicutes, and Actinobacteria [1]. Studies conducted in the North Sea, the coastal Baltic Sea, the North Atlantic, and several freshwater systems revealed that the main bacterial communities on plastics belong to the families of Flavobacteriaceae, Erythrobacteraceae, Hyphomonadaceae, and Rhodobacteraceae [6,9,10]. In addition, a growing body of evidence suggests that microbial dysbiosis in animals is correlated with the toxic effects caused by environmental contaminants [11,12,13,14]. Although there are many concerns about micro- and nanoplastics’ effects on microorganisms, scarce data exist so far, especially for aquaculture surrounding seawater.

Furthermore, today, harmful algal blooms (HABs) represent a frequent phenomenon associated with water eutrophication. HABs can result in damaged aquatic ecosystems, adverse health effects on humans, and important economic damage. Many abiotic factors are implicated in bloom development, e.g., nutrients, temperature, light, and radiation. Apart from the abiotic factors, a significant influence of the bacterial community on algal bloom dynamics has been observed. Moreover, alterations during the blooms led to changes in bacterial population dynamics, and an increase in bacterial abundance associated with several bacterial groups (e.g., *Rhodobacterales*) was observed [15]. Many bacterial strains have been identified in samples derived from diatoms and dinoflagellates, which have thus been characterized as the main phycosphere genus [16]. More specifically, HABs occasionally occur in marine mussel and fish farming areas—probably because of intense eutrophication, with fish farming being more detrimental in terms of causing this phenomenon—and are particularly responsible for economic losses in mussel farming because of toxin accumulation in mussel tissues [17].

The Thermaikos gulf (North Greece) is the most important *Mytilus galloprovincialis* farming area in Greece since 85% to 90% of Greek mussel production is located here, promoting local business activities. A total of 80% of the production, about 30,000 tons per year, is exported to European Union countries, boosting the local economy. Moreover, the Thermaikos gulf attracts tourism, and many recreational activities are observed. Additionally, the deltas of three large rivers, the Loudias, Axios, and Aliakmon, which flow into the gulf, contain a large wetland protected by the Convention of Ramsar that provides great ecological importance to the Thermaikos gulf [18]. However, a plethora of potentially polluting activities take place in the surrounding areas, including agricultural and animal farming enterprises, which may be reflected in the shaping of bacterial communities. Correspondingly, the flow of the aforementioned rivers may occasionally bring urban waste, mainly attributed to Thessaloniki, located in the northern part of the gulf, which is the second-largest city in Greece. Thus, there is an increasing interest in generating new knowledge on bacterial abundance and its diversity in aquaculture water and the surrounding seawater in the coastal area. For instance, *Pseudo-nitzschia* sp. and enterococci have been well described in this area [19,20]. The Thermaikos gulf constitutes a mesotrophic basin, and because of its location close to the urban area of Thessaloniki Bay, it is facing strong anthropogenic pressure both from the city’s harbor and industrial human activity. In addition, the delta formed by four rivers flowing toward the gulf passes through a hydroelectric power dam located in Western Macedonia correlated with high pollution levels [21] The levels of petroleum hydrocarbons and n-alkanes have been found to be increased in the upper layers of the gulf [22]. Furthermore, little is known concerning the microcystin (MC) presence in the Mediterranean mussel *Mytilus galloprovincialis* farming area, and in a recent study, MCs were detected for the first time in the mussel production area in the Thermaikos gulf [18]. In addition, limited data refer to the microbial composition of water samples from aquaculture in the Thermaikos gulf. Most of these studies have reported the bacterial communities detected within the tissues of marine bivalves [17,20,23] as well as in sediments [22,24,25].

Taking into consideration the significant contribution of *Mytilus galloprovincialis* exports to the Greek economy, it is of rising importance to add knowledge concerning the bacterial species presence in water samples from Greek mussel’s aquaculture zones from a public health point of view. Thus, the scope of the current study was the investigation of the bacterial community existing in water samples from *Mytilus galloprovincialis* aquaculture areas in the Thermaikos gulf, northern Greece, that may provoke toxicity in aquatic organisms and humans, environmental pollution in the mussel production area, and algal blooms. Water samples from the mussel aquaculture zones of the Thermaikos gulf were examined to (a) detect and investigate the taxonomic description of local bacteria populations, (b) investigate their phylogenetic relationships with closely related bacteria, and (c) explore possible assumptions about their origin and their effects on organisms and humans.

## 2. Materials and Methods

### 2.1. Sampling Area

The sampling sites included five marine areas, which were within the *Mytilus galloprovincialis* farming area, whereas two sites originated from brackish waters near the estuaries of the deltas of the Axios and Gallikos rivers located in the Thermaikos gulf; in total six sampling sites (Figure 1). The Thermaikos gulf is a semi-closed estuary with a 90 m maximum depth and a surface of 5100 km^2^ that is in the northwest Aegean Sea in Central Macedonia, Greece. The sampling sites included Kavoura Chalastra (40°32′20.12′′ N, 22°44′56.63′′ E), Klidi Imathia (40°28′37.03′′ N, 22°39′58.94′′ E), Makrigialos Pieria (40°24′57.98′′ N, 22°37′14.93′′ E), Aggelochori Thessaloniki (40°29′30.05″ N 22°49′11.79″ E), the delta of the Axios river (40°37′50.85″ N, 22°50′46.13″ E), and estuary of the Gallikos river (40°37′50.85′′ N, 22°50′46.13′′ E) (Figure 1). These sites are all considered burdened owing to agricultural, industrial, veterinary, and urban wastes. However, they remain of high economic and environmental importance.

In total, 755 water sample batches were collected twice or three times per month. Most of the samples were collected during the spring, summer, and autumn (645 out of 755) because of the seasonality of the mussel production period and the potential presence of toxic species during the warm periods of the year. The sampling of marine waters was performed by the water column method using a portable hosepipe sampler. Surface samples of brackish waters were collected using a telescopic water sampler.

In addition, measurements of abiotic parameters of the water (temperature, salinity, pH, and dissolved oxygen) were conducted during samplings using a handheld multiparameter instrument (YSI 556, Xylem Inc., Yellow Springs, OH, USA). For sample transportation, a portable refrigerator (at 4 ± 1 °C) was used, and the samples were transferred promptly to the Laboratory of Microbiology and Infectious Diseases, Faculty of Veterinary Medicine, Aristotle University of Thessaloniki, for further analyses.

### 2.2. Bacteria Isolation and Molecular Identification

A total of 150 mL of each sample was filtered through 0.45 μm pore diameter filters (PALL CORPORATION, 600 South Wagner Road Michigan). Two filters were utilized, with one proceeding to the culture and the other stored at −70 °C. The method for bacterial culture from water samples was applied as described in [26].

Total DNA was isolated from cultures using the QIAamp DNA Mini Kit (Qiagen, Valencia, CA, USA) following the manufacturer’s protocol. The concentration and purity of the isolated DNA were estimated using a NanoDrop spectrophotometer (Shimadzu, Kyoto, Japan). A part of the 16S rRNA in the bacterial genome was amplified using the primers 27f-CM and 1492r [27], which amplify approximately 1400 base pairs, using the MyTaqTM Red Mix (Bioline, London, UK). PCR conditions included an initial desaturation step of 3 min at 95 °C, 38 cycles of 30 s at 95 °C, 40 s at 51 °C, and 50 s at 72 °C, followed by a final extension step of 7 min at 72 °C. The PCR products were checked by electrophoresis in 1.5% agarose gel stained with ethidium bromide, and successfully amplified ones were sequenced to identify the hosting bacteria [28]. The successfully amplified PCR products were purified with the NucleoSpin PCR Clean-up Kit (Macherey-Nagel, Düren, Germany) following the manufacturer’s instructions and sequenced on both strands, applying the Big Dye Terminator v3.1 sequencing method using the forward and reverse primers in an ABI Prism 3730XL automatic capillary sequencer (CeMIA, Larissa, Greece) with both PCR primers. The editing and alignment of individual sequences were performed using the software MEGA version X, applying the MUSCLE algorithm [29]. Since the scope of the phylogenetic analysis was not to evaluate evolution rates but to discriminate the various bacterial species using the pairwise gap deletion and on the basis of p-distances, a neighbor-joining (NJ) tree was constructed using the program MEGA version X, incorporating confidence intervals obtained by 1000-replicate bootstrapping.

## 3. Results

### 3.1. Bacterial Detection

Bacterial cultures indicated the presence of cyanobacteria in 58 out of the 755 examined water samples, whereas axenic cultures were obtained in 22 samples. Moreover, other genera of marine bacteria were observed that were not classified as cyanobacteria according to Anagnostidis and Komárek [30]. Cyanobacteria classified morphologically as *Synechocystis* sp. were chosen for sequencing since they are prone to producing microcystins [31]. Alignment of the sequenced fragments resulted in the read of 976 base pairs, which were further phylogenetically analyzed in comparison with closely related species obtained from GenBank.

### 3.2. Phylogenetic Analysis

Four bacterial genera were identified after Basic Local Alignment Search Tool (BLAST) searches of the sequenced amplified fragments on the NCBI website (https://www.ncbi.nlm.nih.gov/ (accessed on 10 November 2021)), i.e., *Halomonas* sp., *Planococcus* sp., *Sulfitobacter* sp., and *Synechocystis* sp., exhibiting sequence similarities greater than 90% in comparison with congeneric haplotypes in the GenBank database. In particular, two haplotypes (THESS1 and THESS4, Figure 2) were categorized as *Halomonas* sp., very closely related to *Halomonas stevensii* and *Halomonas johnsoniae* (Figure 2), both of which were detected in four out of the six sampling sites (Table 1). On the basis of BLAST results as well as the phylogenetic tree in Figure 2, the *Planococcus* strain (THESS3) sequence was very closely related to *Planococcus maritimus* species and was only detected at the Makrigialos sampling site. On the other hand, neither *Sulfitobacter* sp. nor *Synechocystis* sp. could be further identified at the species level, as depicted in the dendrogram of Figure 2. The *Sulfitobacter* sp. strain was detected in three out of the six sampling sites, i.e., Gallikos, Chalastra, and Axios (Table 1), all of which are located in the inner part of the Thermaikos gulf. Finally, *Synechocystis* sp. (THESS, Figure 2) was detected in the Chalastra, Makrigialos, and Aggelochori sampling sites, where the main mussel farming marine area is located.

## 4. Discussion

To the best of our knowledge, the present study constitutes the first report of the halophilic bacteria *Halomonas* sp., *Sulfitobacter* sp., and *Planococcus* sp. in water samples from the Thermaikos gulf, the major *Mytilus galloprovincialis* aquaculture area in Greece. In addition, in line with a recent study by our lab detecting microcystins in the same marine area [28], bacteria belonging to the *Synechocystis* genus were identified in seawater from the Thermaikos gulf. An investigation conducted from [22] that studied the variability of the sediment bacterial community composition and diversity from sediments in different regions of the Eastern Mediterranean Sea, including the Thermaikos gulf, revealed high richness in the sediment bacterial communities and noteworthy variability in bacterial composition in the different areas. The results revealed that Alpha- and Gammaproteobacteria were observed at high frequencies in most sediments. Another study revealed that in sediments derived from Thermaikos Gulf, the bacterial community included 31.7% Beta- and Gammaproteobacteria, 3.2% Alphaproteobacteria, and 1.6% Firmicutes [25]. In these bacterial groups, some halophilic and moderate halophils were included. It is reported that they can thrive with moderate salt concentrations (3–15% NaCl); however, if the salinity increases to extreme levels, they can occasionally grow as well [32].

The genus *Halomonas*, detected in most of the sampling sites in the Thermaikos gulf in the current study (Table 1), belongs to the Halomonadaceae family in the class Gammaproteobacteria. Various Gammaproteobacteria members have been reported to grow in the presence of toxins and/or pollutants, such as polyunsaturated aldehydes (PUAs), compounds produced and released from diatoms in natural communities [33,34,35]. It should be noted that the members of this genus are characterized by their preference to grow in saline or hypersaline environments [36], and hence this genus is characterized as halophilic or halotolerant [37]. The bacteria included in the *Halomonas* genus are distributed in a wide range of environments in terms of pH, temperature, and salinity [38]. Furthermore, microorganisms in this genus are useful for bioremediation uses on account of the degradation activity of hydrocarbons under hypersaline conditions as well as for wastewater treatment [37,39]. In addition, their extracellular exopolysaccharide production promotes the production of biofilm in marine environments. Biofilm production has been recorded in these species, and this could contribute to their persistence in polluted aquatic environments [40,41,42,43]. There are some cases of infections reported, attributed to *Halomonas* bacteria causing bacteremia [37,44,45,46,47,48], and it was proposed by Kim et al. (2013) [49] that the medical community should be more aware of the pathogenic potential of *Halomonas* bacterial species.

Furthermore, *Halomonas* bacteria have been isolated from coastal marine communities of the chronically polluted Priolo Bay on the eastern coast of Sicily, Italy [50]. In addition, *Halomonas* species were isolated among other bacteria from the Adriatic Sea during a diatom bloom [51] as well as from a shallow pond in Hungary during an algal bloom [52]

Similarly, regarding *Halomonas* bacteria, *Sulfitobacter* sp. was detected in the majority of the sampling sites examined, especially the ones in the inner part of the Thermaikos gulf (Table 1, Figure 1). The inner part of the Thermaikos gulf is characterized by a cyclonic flow that prohibits the renewal of the seawater influencing all aquatic organisms [53]. These results are in line with [54], which detected bacteria belonging to the *Sulfitobacter* genus in mussels, floating particles (FP), biofilm, and water samples from two different sites across Tunisian coastal areas. In various marine regions, the dominance of Proteobacteria in the clone libraries of seawater has been observed, as well in seawater contaminated with the water-soluble fraction (WSF) of crude oil [54,55,56,57,58,59,60]. Most of the Proteobacteria detected in sub-Antarctic seawater samples collected in Ushuaia, Argentina, were members of the family Rhodobacteraceae, closely related to the *Sulfitobacter* genus [61]. There are studies indicating that members of the *Sulfitobacter* genus are mainly sulfite oxidizers and hence have been identified in a plethora of habitats, including the Mediterranean Sea [62] and hypersaline environments [63], starfish, and seagrass [64]. Harmful algal blooms usually caused by water eutrophication and bacterial communities seem to shift during a bloom with Rhodobacterales including dominant bacterial groups [15]. In addition, *Sulfitobacter* members have been isolated during a phytoplankton bloom in the Southern North Sea [65], from a North Atlantic algal bloom [66], from the microbiome of a copepod during an algal bloom in the North Atlantic [67], from surface seawater in the East China Sea during an algal bloom [15], and from Antarctic polynyas, and they are reported to be one of the most abundant clades in eutrophic seas. Consequently, they might be correlated with bacterial bloom as well [68]. Furthermore, many strains of *Sulfitobacter* have been identified from samples of diatoms and more specifically one strain (*Sulfitobacter pseudonitzschiae* H46) has been isolated from dinoflagellates. As a result, these observations lead to the conclusion that the genus *Sulfitobacter* is a phycosphere genus [16]. In addition, *Sulfitobacter* populations were found to be associated with phytoplankton blooms in the spring and summer [69]. Consequently, it can be estimated that *Halomonas* sp. and *Sulfitobacter* sp. detected in the present study might be indicators of some form of pollution in Thermaikos gulf, which is particularly polluted in its inner part [70].

On the other hand, *Planococcus* sp. was only detected in Makrigialos, located in the outer part of the gulf and characterized by greater depths. Bacteria classified in the *Planococcus* genus are Gram-positive and characterized as halophilic, and they are mainly known for their ability of hydrocarbon degradation and biosurfactant/bio emulsifier secretion [71] *Planococcus* species have been isolated from oil-contaminated areas in Iran [72] and from oil-contaminated soil in the Qinghai-Tibetan Plateau [73]. *Planococcus* were additionally detected in high abundance on plastic debris derived from Haihe Estuary, located in Bohai Bay, China [74]. In general, plastic pollution of marine environments is a problem that is gaining more and more attention. Plastics are widely present in marine environments and may exhibit toxic effects on organisms and humans. Apart from aesthetic degradation, plastics, microplastics, and nanoplastics are correlated with bacterial dysbiosis. There is evidence that microplastics from rivers may accumulate in aquaculture waters [3] and that they can provide the microbiota with a place for further development [75,76,77]. In the case of the Thermaikos gulf, the presence of microplastics can be directly correlated with mussel farming. Particularly at the Makrigialos sampling site, *M. galloprovincialis* farms are based on the long line farming system, in which mussels grow within tubular nets made of extruded polyethylene. Each plastic tubular net remains in the seawater for half of the growing period of the mussels, approximately 3–6 months, when the tubular nets are replaced with new ones for the other half of the growing period. It should be also noted that, occasionally, used nets are thrown into the sea instead of being collected to be recycled or destroyed, further burdening the already affected benthos [78].

*Synechocystis* sp. constitutes a potentially toxic cyanobacterium genus that was detected in the main mussel farming area in the Thermaikos gulf (Table 1). Adverse health effects are known to be caused both in human and aquatic organisms by toxic cyanobacteria inhabiting aquatic ecosystems. Toxic compounds with hazardous effects are attributed to a group of secondary metabolites of cyanobacteria known as cyanotoxins [79,80,81]. According to their biological impacts, they can be classified as hepatotoxins (microcystins (MCs) and nodularins), cytotoxins (cylindrospermopsins), or neurotoxins (anatoxins and saxitoxins) [82,83,84]. Among them, microcystins are the most abundant category of cyanotoxins exhibiting high toxicity [5]. Exposure to high doses of cyanotoxins is correlated with respiratory failure and nerve dysfunction, which can lead to death [83,85]. More specifically, severe effects caused by microcystins have been related not only to humans but also to domestic, wild, and aquatic organisms [86]. Their toxicity is reported in many organs, including the kidneys [5], intestines [87], brain [88], heart [89], and lungs [74,90]. The genera of *Synechocystis* sp. isolated from the Thermaikos gulf in the current study produced microcystins during the experimental exposure of Mediterranean mussels *Mytilus galloprovincialis* at levels up to 6.85 ± 0.220 μg/kg after 72 h of exposure at a density of 100,000 cells/mL [31]. More than 150 genera of cyanobacteria have been described so far. Among them, at least 40 are characterized as cyanotoxic or toxin-producing cyanobacteria [91,92]. Among the most studied genera that produce MCs are *Microcystis*, *Anabaena* (*Dolichospermum*), *Nostoc*, *Planktothrix*, and *Chroococcus* [93], and MCs are also produced by marine picoplanktonic species, such as *Synechococcus* and *Synechocystis* [18,94]. Consequently, there is an emerging need for the detection, identification, and monitoring of local cyanobacteria exhibiting toxicity, especially those that are not usually investigated.

## 5. Conclusions

In conclusion, bacterial isolation from water samples originating in the Thermaikos gulf, the major mussel aquaculture marine area in Greece, revealed the presence of the *Sulfitobacter*, *Halomonas*, *Planococcus*, and *Synechocystis* bacterial genera. Members of *Halomonas* and *Sulfitobacter* have been isolated from highly polluted sites and may be correlated with water pollution in the Thermaikos gulf as well. On the other hand, *Planococcus* bacteria have been identified in samples derived directly from plastic debris and may be related to microplastics that are formed on account of plastic net residues utilized in the long line mussel farming system that is applied in the Thermaikos gulf. The detection of *Synechocystis* bacteria is in line with a recent study detecting microcystins in seawater from the Thermaikos gulf [28], suggesting the need for systematic monitoring in mussel farming areas. Overall, our study indicates that investigating bacterial populations dominating in aquaculture water samples, and not only directly investigating the organisms cultivated, is of high importance. In this manner, associations between dominant bacterial genera and harmful algal blooms, pollution indices, and the presence of toxins may be performed.

## Figures and Tables

**Figure 1 ijerph-19-05285-f001:**
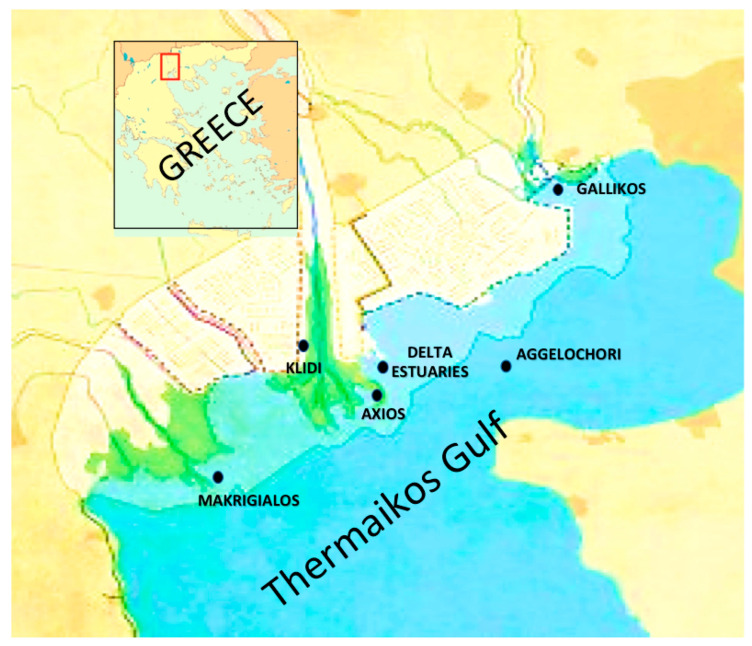
Sampling sites in Thermaikos gulf [18].

**Figure 2 ijerph-19-05285-f002:**
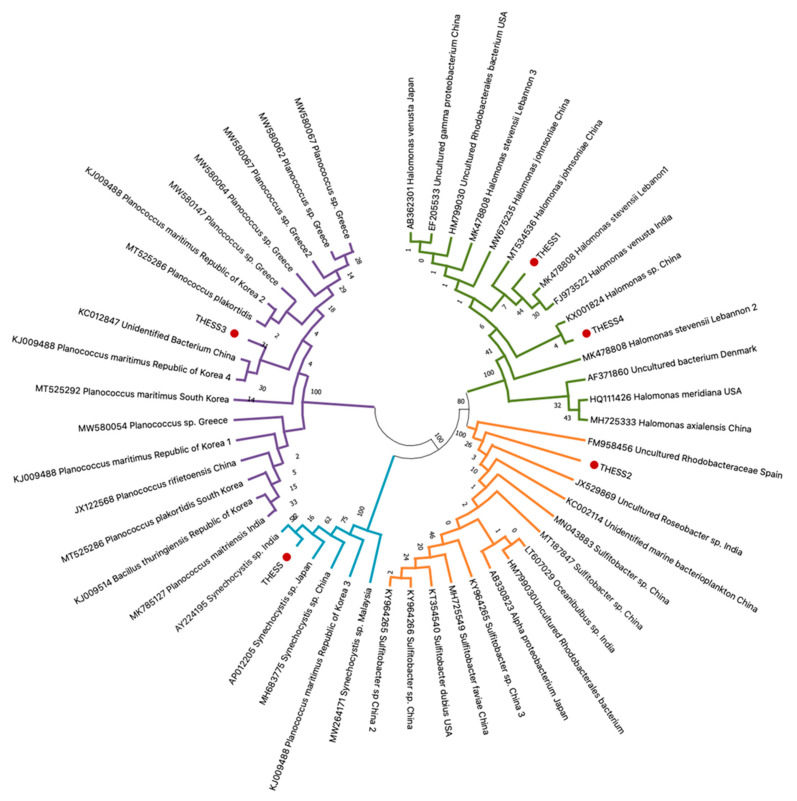
Neighbor-joining (NJ) dendrogram depicting the phylogenetic relationships of the 16S rRNA haplotypes originating from seawater samples from mussel aquaculture in the Thermaikos gulf, with the most closely related congeneric haplotypes available in the GenBank database. Accession number, taxonomic classification, and geographic origin for each haplotype obtained from GenBank are indicated on each branch. Novel sequences derived in the present study are indicated with red dots. Confidence intervals based on 1000 iterations are demonstrated on each clade.

**Table 1 ijerph-19-05285-t001:** Geographical distribution of the detected bacterial genera.

Bacterial Genera	Sampling Areas					
	Gallikos	Kavoura Chalastra	Axios Delta	Klidi Imathia	Makrigialos	Aggelochori
*Halomonas* sp.	+	+	+	+		
*Planococcus* sp.					+	
*Sulfitobacter* sp.	+	+	+			
*Synechocystis* sp.		+			+	+

(+): bacteria detected.

## Data Availability

All data are included within the manuscript or public databases.

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
