# Peer review of "Pollution Indicators and HAB-Associated Halophilic Bacteria Alongside Harmful Cyanobacteria in the Largest Mussel Cultivation Area in Greece"

_ijerph, 2022, doi:10.3390/ijerph19095285_

Round 1
Reviewer 1 Report
The new version improves a lot compared to the first one. After checking the new input, I would like to give green light to the acceptance.
Author Response
We would like to thank the reviewer for for recognising the improvement of the new version of our manuscript
Reviewer 2 Report
The authors have improved the manuscript in this novel version.
I still have additional comments:
Line 87-the sentence «HABs have often been related to marine mussel farming areas» seems to imply mussel farming causes HABs. Fish farming is much more detrimental in causing eutrophication, than mussel farming. HABs cause economical losses in mussel farming due to toxin accumulation in mussels. HABs are natural phenomena, and mussel farming is not necessarily correlated with their appearance.
Line 157-the samples were collected ‘bi-monthly’ (2x times) or 3x times per month. ‘Monthly’ was not the rule, maybe an exception.
Line 162- « method of water column» was not described properly. Was this a ‘hosepipe sampler’, to collect an integrated water column sample?
Line 207- the use of ‘although’ does not seem necessary in the context.
Line 214-«in comparison closely related species» ‘with’ seems to be missing here.
Line 232-the description of five sampling sites seems in contradiction with Fig. 1 or Table 12, where I can see 6 sampling sites.
Line 387- « at least 40 are characterized as [91,92].» ‘Were characterized by 91,92’? Sentence does not make sence.
Author Response
We thank the reviewer for finding improved. Taking into consideration the suggested additional comments the following modifications were performed:
Line 87-the sentence «HABs have often been related to marine mussel farming areas» seems to imply mussel farming causes HABs. Fish farming is much more detrimental in causing eutrophication, than mussel farming. HABs cause economical losses in mussel farming due to toxin accumulation in mussels. HABs are natural phenomena, and mussel farming is not necessarily correlated with their appearance.
Response: The sentence was rephrased as follows: "More specifically, HABs occasionally occur in marine mussel and fish farming areas, probably due to intense eutrophication with fish farming beingmore detrimental in causing this phenomenon,and are particularly responsible for economical losses in mussel farming due to toxin accumulation in mussel tissues", as recommended by the reviewer
Line 157-the samples were collected ‘bi-monthly’ (2x times) or 3x times per month. ‘Monthly’ was not the rule, maybe an exception.
Re: The sentence was rephrased as follows: "In total 755 water sample batches were collected twice or three times per month" for clarity
Line 162- « method of water column» was not described properly. Was this a ‘hosepipe sampler’, to collect an integrated water column sample?
Re: Indeed this was referring to a hosepipe sampler, which has been corrected in the revised manuscript
Line 207- the use of ‘although’ does not seem necessary in the context.
Re: "although" was replaced by "whereas" in accordance to the reviewer's comment
Line 214-«in comparison closely related species» ‘with’ seems to be missing here.
Re: "with" was added as recommended by the reviewer
Line 232-the description of five sampling sites seems in contradiction with Fig. 1 or Table 12, where I can see 6 sampling sites.
Re: "five" was replaced by "six" as correctly mentioned by the reviewer
Line 387- « at least 40 are characterized as [91,92].» ‘Were characterized by 91,92’? Sentence does not make sence.
Re: The sentence was corrected as follows: "Among them, at least 40 are characterized as cyanotoxic or toxin producing cyanobacteria" as correctly mentioned by the reviewer
This manuscript is a resubmission of an earlier submission. The following is a list of the peer review reports and author responses from that submission.
Round 1
Reviewer 1 Report
The paper deals with the investigation of microorganism’ community living in water samples where there are six aquacultures of Mytilus galloprovincialis, i.e. mussels commonly used as food. The authors showed as these bacteria are present in the water and not directly in the mussels. Because the possible toxicity that bacteria, producing toxins, can introduce in the food chain, I think that this study is very important for the community and stakeholders.
The paper is relevant in its field; authors state that this is the first report about this, and it is well done.
I suggest publishing it in the present form.
As minor suggestions: fig 1 could be more attractive. Maybe colored?
Reviewer 2 Report
In the manuscript, the authors analyzed the bacterial community existing in water samples from six Mytilus galloprovincialis aquaculture areas in Thermaikos Gulf, and illustrated its relationship with microplastics presence and algal blooms in the water. Generally speaking the content meets well the topic of the journal, and its writing style is OK for the publication. My personal perspective toward this research, however, is that the methods used is not so innovative and the sample size is a bit small. Based on the reasons above, my recommendation is Major Revision. The details are shown below:
Line 5: there is no need to say this is a research related to "Literature Review", since in the discussion it is a must to compare your results with those from other scientists. Better to remove subheading in the title.
Line 59: Better to change "microorganisms" with "bacteria", since in the whole manuscript you talk about the bacteria changes, of which the target is much smaller than the whole "microorganisms".
Line 89: I did not get the causal relationship between "the most important Mytilus galloprovincialis farming area" with "interet on generating new knowledge on bacterial anundance". Could you please explain the reason more clearly??
Line 125: Some important information is missing in the Materials and Methods: you did the sampling at which season, and at which frequency (i.e. the temporary-spatial resolution of samplings)?
Line 189: delete "also"
Line 196: Please firstly indicate the full name of BLAST (i.e. Basic Local Alignment Search Tool) and then give the abbreviation.
Reviewer 3 Report
In this article, the distinction between a regular contribution and a review is quite confusing. I cannot recommend its acceptance.
Detailed comments:
Line 59: ‘from microorganisms’ is not adequate here, maybe ‘by microorganisms’
Line 93: correct to ‘Pseudo-nitzschia’
Line 104: « recent study [18] MCs were…» It is best to put the references at the end of the sentence, or before a comma. Please check throughout the manuscript.
Line 118: check if ‘area’ is missing after ‘production’
Line 142: the figure has low resolution, and sampling site names are not readable.
Lines 186/Section 3.1: the paper seems a regular article, where results are expected. But here, we were presented with a literature review. This was best explained in the introduction. / Next this part is very confusing: « which were further phylogenetically analyzed.3.2. Figures, Tables and Schemes»
Table 1 shows the presence/absence of a few selected genera. But what about the remaining genera?
A study cited in the discussion refers that in sediments, the bacterial community constituted «of 31.7% from beta-Gamma-Proteobacteria, 3.2% Alphaproteobacteria and 1.6% Firmicutes…» The present study does not discriminate any proportions whatsoever. It presents only a very limited set of data. Why only these 4 genera were targeted? I believe this should have been partially explained in the introduction.